# Eco-Friendly Preparation of Carbon-Bonded Carbon Fiber Based on Glucose-Polyacrylamide Hydrogel Derived Carbon as Binder

**DOI:** 10.3390/nano13061045

**Published:** 2023-03-14

**Authors:** Chen Zeng, Yanju Gu, You Xie, Weiqin Hu, Min Huang, Gen Liao, Jianxiao Yang, Zheqiong Fan, Ruixuan Tan

**Affiliations:** 1Hunan Province Key Laboratory for Advanced Carbon Materials and Applied Technology, College of Materials Science and Engineering, Hunan University, Changsha 410082, China; 2School of Materials Science and Engineering, Changsha University of Science and Technology, Changsha 410114, China

**Keywords:** carbon-bonded carbon fiber composites, glucose-polyacrylamide hydrogel, chopped carbon fiber, extrusion-injection molding, thermal conductivity

## Abstract

Lightweight, high-temperature-resistant carbon-bonded carbon fiber (CBCF) composites with excellent thermal insulation properties are desirable materials for thermal protection systems in military and aerospace applications. Here, glucose was introduced into the polyacrylamide hydrogel to form the glucose-polyacrylamide (Glu-PAM) hydrogel. The CBCF composites were prepared using the Glu-PAM hydrogel as a brand-new binder, and the synergistic effect between glucose and acrylamide was investigated. The results showed the Glu-PAM hydrogel could limit the foaming of glucose and enhance the carbon yield of glucose. Meanwhile, the dopamine-modified chopped carbon fiber could be uniformly mixed by high-speed shearing to form a slurry with the Glu-PAM hydrogel. Finally, the slurry was successfully extruded and molded to prepare CBCF composites with a density of 0.158~0.390 g cm^−3^ and excellent thermal insulation performance and good mechanical properties. The compressive strength of CBCF composites with a density of 0.158 g cm^−3^ in the Z direction is 0.18 MPa, and the thermal conductivity in the Z direction at 25 °C and 1200 °C is 0.10 W m^−1^ k^−1^ and 0.20 W m^−1^ k^−1^, respectively. This study provided an efficient, environment-friendly, and cost-effective strategy for the preparation of CBCF composites.

## 1. Introduction

The carbon-bonded carbon fiber (CBCF) composite is a low-density C/C composite with chopped carbon fiber as the skeleton and phenolic resin as the binder. Due to its low density (0.10~1 g cm^−3^), high porosity (60~90%), low thermal conductivity (<1 W m^−1^ k^−1^), and excellent high-temperature stability, it is widely used in the field of thermal insulation [1,2,3,4,5,6]. At present, the main preparation methods of CBCF composites are vacuum filtration, pressure filtration, and whole needle-punching method [7,8,9,10]. The vacuum filtration method or pressure filtration method has been developed because of its simplicity and efficiency. Liu et al. [11] prepared CBCF composites with a density of 0.23 g cm^−3^ by the pressure filtration method. The thermal conductivity of CBCF composites in the Z direction at 1300 °C is only 0.35 W m^−1^ k^−1^ with excellent thermal insulation performance. However, CBCF composites prepared by this method lead to uneven density distribution and obvious delamination. The whole needle-punching method is an ideal method for the preparation of CBCF composites. Cheng et al. [12] impregnated the whole needle-punched felt with phenolic resin aerogel and obtained CBCF composites with a density between 0.27~0.37 g cm^−3^ and good overall uniformity. However, this method is complicated and requires a long cycle.

In recent years, extrusion-injection molding technology of carbon-fiber-reinforced polymer composites has become the focus of research interest due to its advantages of low cost and flexible design [13,14,15,16,17]. Martin et al. [18] prepared oriented chopped carbon-fiber-reinforced polypropylene composites with anisotropic mechanical properties and thermal conductivity using extrusion additive manufacturing. Nashat et al. [19] prepared high-strength chopped carbon-fiber-reinforced epoxy composites by direct ink writing (DIW) to make the short-cut carbon fibers oriented. Unfortunately, the extrusion-injection molding technology requires a paste with good viscoelasticity, and it is difficult to form a paste with good viscoelasticity with the current phenolic resin binder after mixing with short-cut carbon fibers. Moreover, phenolic resin as a binder will cause environmental pollution, which is contrary to the current development concept of global green environmental protection.

For decades, green renewable carbon precursors have been regarded as one of the most ideal carbon precursors [20,21,22]. Compared with resins, sugars are cheaper, more environment-friendly, and easier to purify. However, due to the instability of small sugar molecules at high temperatures, they are easy to foam to a porous fluffy structure and low carbon yield, which limits their applications in the field of carbon materials. In order to limit the foaming of carbohydrates and increase their carbon yield, researchers have made many attempts, such as the hydrothermal method [23], hard/soft template method [24], and sol-gel method [25,26]. Among them, the sol-gel method has been widely used because of its low cost, simplicity, and high carbon yield. However, at present, the sugar–hydrogel is mainly used as a carbon source to prepare some porous carbon [27], carbon aerogel [28], and carbon film [29]. However, there is no relevant literature report on the use of sugar–hydrogel as a binder for CBCF composites. Fortunately, we found that the syrup-like liquid state can be formed by breaking and diluting the sugar–hydrogel with water. This sugar–hydrogel solution has good rheological properties and viscoelasticity.

In this regard, we propose a novel and simple method to prepare a sugar–hydrogel binder with good rheological properties as well as viscoelasticity by the high-speed crushing of glucose-polyacrylamide (Glu-PAM) hydrogel with water. At the same time, dopamine, a green modification strategy based on bionics, was used to improve the hydrophilicity of carbon fiber and the adsorption of the Glu-PAM hydrogel binder [30]. Then, the uniform mixing of the carbon fiber and binder was realized by high-speed shearing equipment, and a slurry suitable for extrusion molding was formed. Meanwhile, the mechanical and thermal properties of CBCF composites prepared by the extrusion-injection molding method are systematically studied. Compared with the current commercial CBCF insulation composites, the preparation process is simpler and has better thermal insulation performance. It has broad application prospects in the field of thermal insulation.

## 2. Experiments

### 2.1. Preparation of Glucose-Acrylamide Hydrogel Binder

Anhydrous glucose (10~80 g) and *N*,*N*-Methylenebisacrylamide Bis-acrylamide (1 g) were added into 100 mL water, respectively, and heated in water bath at 70 ℃ until they were completely dissolved. Then, 20 g acrylamide (AM), 1 mL initiator ascorbic acid (3%), and 1 mL initiator hydrogen peroxide (3%) were added into the solution. The mixture was stirred with glass rod for 1 min and heated in water bath at 70 °C to obtain Glu-PAM hydrogel. Then, the Glu-PAM hydrogel was broken by a high-speed grinder for 1 min, and then 100 mL water was added to continue the crushing for 1 min to obtain a Glu-PAM hydrogel binder.

### 2.2. Preparation of Dopamine-Modified Carbon Fiber

A total of 10 g of viscose-based chopped carbon fiber (CF) with a length of 3~5 mm was added to 1000 mL of water and sonicated for 30 min. Then, TRIS buffer was added to adjust the pH of the solution to 8.5, and 1 g of dopamine hydrochloride was added. The entire mixture system was mechanically stirred at 25 °C for 24 h. After the reaction, the system water was filtered out by a vacuum filtration device, rinsed with deionized water several times, and finally placed in an oven at 50 °C for 24 h.

### 2.3. Preparation of CBCF Composite Slurry and Molding of CBCF Composites

The 10 g pDop-CF and Glu-PAM hydrogel binder with different glucose contents were mixed with a high-speed shear device at 3000 r/min for 10 min, and the mixed slurry was extruded into a metal mold with a self-made extrusion device (as shown in Figure 1c). The mold was placed in an oven at 120 °C for 12 h, and the free water in the system was removed. Then, we continue to dry it in the oven at 180 °C for 12 h, so that the sample was completely cured and demoulded. Finally, the sample was placed in a tube furnace in an N_2_ atmosphere at 10 °C/min. The temperature was raised to 1200 °C for 2 h. Thus, CBCF composites with different densities of 0.158, 0.238, 0.312, and 0.390 g cm^−3^ were obtained and named as CBCF-1, CBCF-2, CBCF-3, and CBCF-4, respectively. The complete preparation of CBCF composites is shown in Figure 1a. In addition, CBCF commercial insulation composites prepared by vacuum filtration method and laminated bonding method were used as a comparison with CBCF-1, both with densities of 0.183 and 0.211 g cm^−3^, respectively, and named CBCF-5 and CBCF-6.

### 2.4. Characterization

The microstructure of the CBCF composites with different densities and the surface morphology of carbon fiber were observed by scanning electron microscope (SEM, JSM-7610FPlus, Akishima, Tokyo, Japan). Thermogravimetric analyzer (TGA, NETZSCH STA 449, Selb, Germany) was performed at a heating rate of 10 °C/min in N_2_ atmosphere. The volatile substances produced during the heat treatment of the sample were analyzed by TG-MS (Thermo plus EVO2/Thermo Mass Photo, Selb, Bavaria, Germany). The crystal phase structure was studied by X-ray powder diffraction using CuKa radiation (λ =1.5406 Å) on a TD-3300 diffractometer (DanDong, Liaoning, China). The Raman spectrum was analyzed by Renishaw 1000 B device (Waltham, MA, USA), and the excitation laser was 532 nm. The optical contact angle measuring device (Dataphysics OCA20, Filderstadt, Germany) characterizes the wettability of the carbon fiber surface. CBCF-1, CBCF-5, and CBCF-6 were processed into 20 × 20 × 30 mm^3^ cubes and placed horizontally on a 350 °C heating table. The infrared thermal image was collected by Fluke Ti450 infrared thermal imager (Everett, WA, USA) and the surface temperature change of the sample was recorded. A laser thermal conductivity meter (NETZSCH LFA 427, Selb, Bavaria, Germany) was used to measure the thermal conductivity of the sample. The compressive strength was measured on an electronic universal testing machine (Instron 5969, Norwood, MA, USA) at a rate of 0.5 mm/min.

## 3. Results

### 3.1. Formation and Pyrolysis Mechanism of Glu-PAM Hydrogels

In order to analyze the formation mechanism of the Glu-PAM hydrogels’ high carbon yield, glucose, PAM hydrogels, and Glu-PAM hydrogels were tested by thermogravimetric (TG), and thermogravimetric–mass spectrometry (TG-MS). Before testing, the PAM hydrogel and Glu-PAM hydrogel samples were first treated at 100 °C for 24 h to achieve the purpose of removing free water from the system. Figure 2a shows the TG curves, and it can be seen that the carbon yield of the Glu-PAM hydrogel reached 33.5%, which was significantly improved compared with 12.8% of the glucose powder and 19.2% of the PAM hydrogel. The DTG curves of glucose, PAM hydrogels, and Glu-PAM hydrogels showed two main weight-loss temperature intervals as seen in Figure 2b. In the first weight-loss temperature interval (180~240 °C), glucose, PAM hydrogels, and Glu-PAM hydrogels all reached the fastest weight-loss rate near 225 °C, and the difference between the weight-loss rate of the Glu-PAM hydrogels and the mass-loss rate between glucose and the PAM hydrogels in this weight-loss temperature interval was small. In the second weight-loss temperature range (280~420 °C), the fastest weight-loss rates of glucose, PAM hydrogels, and Glu-PAM hydrogels occurred at 314, 405, and 390 °C, respectively, with the weight loss of Glu-PAM hydrogels significantly lagging behind that of glucose. In addition, in this weight-loss temperature range, the weight-loss rate of the Glu-PAM hydrogels was much smaller than that of glucose and PAM hydrogels. Figure 2c shows the TG-MS *m*/*z* = 44 curve, which shows that the Glu-PAM hydrogel produces CO_2_ at a significantly higher temperature than that of glucose, and produces significantly less CO_2_ than glucose. The results of DTG and TG-MS suggest that the high carbon yield of Glu-PAM hydrogels must be related to the ability of the PAM hydrogels to reduce the escape of CO_2_ and other carbonaceous gases from glucose in the second weight-loss temperature range (280–420 °C).

To further analyze the formation mechanism of the high carbon yield in Glu-PAM hydrogels, FTIR tests were performed on samples of glucose and Glu-PAM hydrogels after heat treatment at 180 and 280 °C, respectively, to analyze the structural changes that occurred during their simultaneous mass changes in the carbonization process. From Figure 2d, it can be seen that 3200–3650 cm^−1^ corresponds to the O–H/N–H stretching vibration, 2900–3000 cm^−1^ corresponds to the C–H/C–N stretching vibration, 1620–1680 cm^−1^ corresponds to the C=C/C=O stretching vibration, 900–1300 cm^−1^ corresponds to the C–O/C–C stretching vibration, and 1300–1500 cm^−1^ corresponds to the C–H/O–H bending vibration. It can be seen that the absorption intensity of glucose at C–O/C–C was significantly greater than that of the absorption peak at C=C/C=O at both the heat treatment temperatures of 180 °C and 280 °C, while the Glu–PAM hydrogels had an opposite trend. This indicates that the glucose molecules are more easily converted to substances containing unsaturated C=C bonds rather than decomposed into carbonaceous gases during the heat treatment in the PAM hydrogel system. It can effectively reduce the loss of carbon during the heat treatment of glucose.

The formation mechanism of the high carbon yield of the Glu-PAM hydrogel can be derived by combining the results of the TG, TG-MS, and FTIR tests. In the process of high temperature pyrolysis, glucose will produce a large number of volatile substances due to the caramelization reaction, such as gaseous water, carbon dioxide, furan derivatives, ketones, etc. The production of these volatile substances leads to the uncontrollable foaming of glucose, forming a porous fluffy structure, thereby reducing the carbon yield of glucose. In contrast, the three-dimensional network structure of the PAM hydrogel will undergo uneven shrinkage during drying dehydration, resulting in the final structural deformation. Glucose was introduced into the PAM hydrogel system; glucose is uniformly dispersed in the PAM hydrogel. Glucose and PAM are combined by hydrogen bonding (the reaction in the gel process is shown in Figure 1b), and there will be a synergistic effect between the two. On the one hand, glucose can be immobilized in situ in a three-dimensional network of PAM hydrogels. On the other hand, during the dehydration process of the Glu-PAM hydrogel, glucose will play a role in supporting the three-dimensional network structure of the PAM hydrogel and limiting its shrinkage. This will also increase the stress in the PAM hydrogel network, while glucose in the PAM hydrogel system exhibits more dehydration condensation conversion to ketones, aldehydes, and unsaturated C=C structures at lower temperatures (180~280 °C), which are more difficult to decompose and volatilize at higher temperatures due to the high-pressure environment provided by the three-dimensional network structure of PAM. At the same time, these high concentrations of unsaturated structures will be more likely to polymerize into polymer structures with a higher thermal stability rather than forming carbonaceous gases such as CO_2_, etc. This is not only beneficial for reducing the loss of carbon during the pyrolysis of glucose in the second weight-loss temperature interval (280~420 °C), but also for facilitating and ensuring the structural integrity.

### 3.2. Characterization of Dopamine-Modified Carbon Fiber

Figure 3a shows the Raman spectra of carbon fiber before and after modification, and there are obvious D and G peaks at 1360 cm^−1^ and 1580 cm^−1^. The D peak is related to the amorphous structure of carbon and the G peak is related to the graphitized structure of carbon. It can be seen that the D peak of CF is obviously enhanced after dopamine modification, and the I_D_/I_G_ of the modified CF is calculated to be 1.05, and the I_D_/I_G_ of the original CF is 0.79. This indicates that the surface of CF is covered by other amorphous carbon. The FTIR test results are shown in Figure 3b, and it can be observed that pDop-CF has more obvious peaks at 3200–3400 cm^−1^ and 1300 cm^−1^, which originated from the N-H stretching vibration of polydopamine and the shear vibration of the C–O group, respectively. The elemental species as well as the relative content of the CF surface were characterized by XPS. The spectral peak patterns of pDop-CF and CF in Figure 3c are similar, but the intensity of the N1s at 399.8 eV and the O1s photoelectron peak at 532.5 eV of the pDop-CF are clearly enhanced compared to the original CF, and the relative content of the O and N elements is obviously increased, which is certainly related to the encapsulation of polydopamine on the CF surface. The C1s of pristine CF as well as pDop-CF were separated into peaks, and the results are shown in Figure 3d,e. The C1s pattern of the original CF consists of C–C/C=C at 284.6 eV, C–O at 286.6 eV, and O=C–O at 289.0 eV. While a new signal appeared in the C1s profile of pDop-CF, the C–N formation structure is located at 285.8 eV, which originated from the C–N structure in polydopamine. Figure 3f shows the N1s profile of pDop-CF, and the N1s peaks appearing at 399.0, 399.5, and 401.0 eV correspond to the –NH_2_, –NH–, and C–N structures, respectively. These results indicate that the modification of the polydopamine coating on the surface of CF is successful.

In order to further analyze the morphological state of pDop-CF, the surface morphology and hydrophilicity of CF before and after dopamine modification were characterized by SEM and the static surface contact angle. From Figure 4a,b, it can be seen that the surface of pDop-CF is smoother compared to the original CF, and the presence of uniformly coated polydopamine films on the surface is obvious. The presence of the polydopamine film changes the surface wettability of CF. The surface contact angle test results are shown in Figure 4c,d. The surface contact angle between CF and water is 128.3°, while the contact angle between pDop-CF and water becomes 88.9°, which is a significant decrease compared with that after dopamine modification, indicating that the hydrophilicity of carbon fiber is improved after dopamine modification. This is mainly attributed to the fact that there are more active structures (such as amino and phenolic hydroxyl groups) in the polydopamine structure coated on the surface of carbon fiber [30], and the reactive groups on the surface easily interact with hydroxyl groups in water to improve the hydrophilicity of the chopped CF.

### 3.3. Microstructure of CBCF Composites

Figure 5 shows the microstructure of the CBCF composites with different densities, and it can be seen that the composites prepared by the extrusion-injection molding method conforms to the structure of typical CBCF composites. From Figure 5a–d, it can be seen that the chopped CFs are randomly arranged along the XY direction. From Figure 5e–h, it can be seen that the chopped CFs have an obvious layered structure in the Z direction, and the CBCF composites have distinct anisotropy in the XY direction and the Z direction. This special microstructure determines that the CBCF composites have special mechanical and thermal physical properties. Figure 5i–l shows the high magnification SEM image of CBCF-1. The adjacent chopped CF inside the CBCF composites is bonded by glucose-cracked carbon at the intersection point, which constitutes a three-dimensional network structure with interconnected fibers. The glucose-cracked carbon is mainly distributed in the laps between the fibers, but there is also some cracked carbon encapsulated on the CF, which further strengthens the bonding force and makes the CBCF composite material have a stable three-dimensional structure. This particular carbon-bonded carbon fiber structure is formed during the curing process of Glu-PAM hydrogel; the Glu-PAM hydrogel flows along the carbon fiber surface and eventually collects at the intersection between neighboring fibers. In addition, a small portion of the Glu-PAM hydrogel continues to remain on the fiber surface wrapping the fibers to form a Glu-PAM hydrogel film layer, a special structure that is retained after curing and carbonization.

### 3.4. Mechanical Properties of CBCF Composites

In order to analyze the effect of the anisotropic microstructure on the mechanical properties of the CBCF composites, the compressive properties of four different densities of the CBCF composites were tested in the XY direction and Z direction. Figure 6a,b shows the compressive stress–strain curves of CBCF-1, CBCF-2, CBCF-3, and CBCF-4 in the XY and Z directions, respectively. The curves can be divided into two stages: The first stage is the elastic deformation stage, and the stress in this stage grows linearly to the maximum elastic stress (point A), which is defined as the compressive strength of CBCF composites. It can be seen that when the density of the CBCF composites increases from 0.158 to 0.390 g cm^−3^, its compressive strength in the XY and Z directions increases from 0.30 MPa to 1.95 MPa in the XY direction and 0.18 MPa to 0.83 MPa in the Z direction, respectively, and the compressive strength in the XY direction is obviously greater than that in the Z direction. This is mainly attributed to most of the fibers in the Z direction being perpendicular to the compression stress direction, while the main deformation mode in the elastic deformation stage is caused by the bending and rotation of the short-cut carbon fibers. The fibers in the Z direction hardly play a role in bearing stress, and plastic deformation will occur at a smaller stress, so the compressive strength in the Z direction is smaller compared to the XY direction. In addition, the compressive strength of CBCF had a trend of increasing with the density of the CBCF composites. This is attributed to the fact that, as the density of the CBCF composites becomes higher, the glucose-cracked carbon located in the internal pores of the CBCF composites and coated on the fiber surface also increases. However, the cracked carbon could restrict the bending and rotation of the fibers, thus increasing the elastic deformation capacity of the fibers.

After the elastic deformation stage comes to the second stage of plastic deformation, it can be seen that, from point A, the CBCF composites display two different deformation response mechanisms in the XY and Z directions; the stress in the XY direction undergoes a small rapid drop and then slowly stabilizes, followed by stress fluctuations in an approximately horizontal and unstable manner until the specimen develops macroscopic fracture. This compression behavior is typical of the initial elastic deformation followed by continuous loading, producing permanent damage within the material, including irreversible internal fiber buckling, and inter-fiber and carbon-bond-point damage. While in the Z direction, the stress exhibits a continued slow increase with increasing strain, this is mainly attributed to the fact that most of the fibers are perpendicular to the compressive stress loading direction. The compressive load can redistribute within the CBCF composites through fiber transfer; the materials are continuously compressed and densified, and is able to absorb more external energy.

### 3.5. Thermal Properties of CBCF Composites

Figure 7a shows the thermal conductivity of the CBCF composites with different densities in the XY direction and Z direction at room temperature 25 °C. It can be seen that CBCF-1 has a low thermal conductivity at room temperature with 0.15 W m^−1^ k^−1^ and 0.10 W m^−1^ k^−1^ in the XY direction and Z direction, respectively. As we all know, the thermal conductivity of insulation materials mainly consists of solid thermal conductivity, gas thermal conductivity, and radiation thermal conductivity [31,32,33]. At room temperature, the solid thermal conductivity has a great influence on the thermal conductivity of the CBCF composites, and the thermal conductivity of the solid phase relies on the energy transfer from the microscopic masses of matter. Therefore, the solid-phase thermal conductivity of the CBCF composites is mainly related to the thermal conductivity of its skeleton material and the skeleton density of the material, while CBCF-1 is mainly composed of viscose-based carbon fibers and a small portion of glucose-cracked carbon. Figure 7b shows the XRD patterns of viscose-based carbon fiber, glucose-cracked carbon, and CBCF-1. There are obvious diffraction peaks at 24° and 43°, respectively. The strong diffraction peak at 24° is attributed to the (002) crystalline plane of graphite and the diffraction peak at 43° is attributed to the (110) crystalline plane of disordered carbon material. The (002) crystalline spacings of the three samples calculated using the Bragg formula are 0.3695 nm, 0.3676 nm, and 0.3649 nm, respectively. It is obvious that the graphitization degree of rayon-based carbon fibers, glucose-cracked carbon, and CBCF-1 is very low, which is typical of hard-to-graphitize carbon. Therefore, the CBCF composites exhibited excellent thermal insulation performance. Figure 7d shows the results of the piezometer test of CBCF-1, indicating that CBCF-1 has a porosity of 89.83%, which indicates that the carbon fibers of the whole CBCF composites have little contact with each other, and the material has few solid-phase heat transfer paths. Therefore, CBCF-1 has a very low thermal conductivity at room temperature.

Figure 7a also shows that as the skeletal density of the CBCF composites increases from 0.158 to 0.430 g cm^−3^, the room temperature thermal conductivity of the CBCF composites increases from 0.10 W m^−1^ k^−1^ to 0.29 W m^−1^ k^−1^ in the XY direction and 0.17 W m^−1^ k^−1^ to 0.49 W m^−1^ k^−1^ in the Z directions, respectively. This is mainly attributed to the fact that, as the skeleton density of the CBCF composites increases, the porosity of the CBCF composites decreases and the carbon fibers are in more contact with each other, resulting in more solid-phase heat transfer paths and an increase in solid-phase thermal conductivity. In addition, it can be clearly observed that the thermal conductivity of the CBCF composites in the XY direction is significantly larger than that in the Z direction. This is mainly attributed to the fact that the short-cut carbon fibers in the Z direction form a quasi-laminar structure perpendicular to the heat flow direction, and the existence of junctions between different layers enhances the thermal resistance in the Z direction, while in the XY plane, a continuous carbon fiber network structure is formed between different short-cut carbon fibers, which enhances the heat transfer in the long path.

Figure 7c shows the thermal conductivity of CBCF-1 from 25 °C to 1200 °C. As the temperature increases from room temperature to 1200 °C, the thermal conductivity of CBCF-1 increases from 0.10 W m^−1^ k^−1^ to 0.20 W m^−1^ k^−1^ in the XY direction and 0.15 W m^−1^ k^−1^ to 0.27 W m^−1^ k^−1^ in Z direction, respectively. This is because the thermal conductivity of gases and radiation gradually become the main influencing factors, the heat transfer of gases is mainly through the heat conduction of gas molecules, and according to the molecular motion and collision theory [34], the heat transfer of gas molecules is through the mutual collision of fast-moving molecules at the high-temperature end and molecules at the low-temperature side with a smaller rate of motion and energy transfer. The larger the average value of the distance moved by gas molecules, the better the thermal conductivity; i.e., the gas thermal conductivity of the CBCF composites is related to its pore size. Figure 7d shows the pore-size distribution of CBCF-1, and its pore size is mainly distributed in 50–150 μm; these larger pore sizes increase the collision frequency of air molecules at high temperatures, and gas molecules collide with each other in these large pores, thus increasing the gas thermal conductivity of the CBCF composites. According to the Rossland theory [35], the radiation thermal conductivity of the thermal insulation material is proportional to the cubic of temperature. Therefore, as the temperature increases, the radiation thermal conductivity of the CBCF composite material will increase sharply, especially above 1000 °C. The radiative thermal conductivity becomes the main factor affecting the high-temperature thermal conductivity of the CBCF composites, resulting in a large increase in the thermal conductivity of the CBCF composite material above 1000 °C relative to room temperature.

In order to evaluate the thermal insulation effect of CBCF-1 prepared by the extrusion-injection molding technique, the variation of surface temperature of the CBCF composites with heating time was recorded by infrared thermography. In addition, the commercial CBCF insulation composites CBCF-5 and CBCF-6 were prepared by the vacuum filtration method, and the laminated bonding method was used as a comparison. The surface temperature changes of the three CBCF composites during the heating process are recorded in Figure 8a, and it was observed that the surface temperature of CBCF-1 was only 51.9 °C after 90 s of heating due to its very low thermal conductivity, which is significantly lower than CBCF-5 (121.5 °C) and CBCF-6 (81.5 °C); CBCF-1 has shown an excellent thermal insulation capability. In addition, the IR thermography test results visually show the difference in temperature distribution between the three CBCF composites prepared by different molding methods during the heating process (Figure 8b); CBCF-1 shows the slowest temperature response during heating, transferring the accumulated heat at a slower rate, and its surface exhibits a bright background color. In short, compared with the commercial CBCF thermal insulation composites prepared by the laminated curing method or vacuum filtration method, the CBCF composites prepared by the extrusion-injection molding technology exhibited better thermal insulation properties. This indicates that the extrusion-injection method is one of the extremely promising CBCF-composite preparation methods.

## 4. Conclusions

In this work, we used the synergistic interaction between glucose and polyacrylamide hydrogel to limit the foaming of glucose, which greatly improved the carbon yield of glucose, and successfully used it as a binder for CBCF composites. The Glu-PAM hydrogel was broken by adding water to form a liquid Glu-PAM hydrogel binder with good viscoelasticity. Then, the liquid Glu-PAM hydrogel binder and dopamine-modified chopped carbon fiber with good hydrophilicity were mixed by high-speed shear mixing to form a slurry with good rheological properties. Finally, the slurry was extruded-injected to obtain the CBCF composites. The chopped carbon fibers of the CBCF composites prepared by this method exhibit a distinct quasi-layered structure in the Z direction, resulting in CBCF-1 exhibiting extremely low thermal conductivity in the Z direction (25 °C & 0.10 W m^−1^ k^−1^, 1200 °C & 0.20 W m^−1^ k^−1^). In addition, due to the high carbon yield of the Glu-PAM hydrogel and the uniform distribution of pyrolytic carbon, the mechanical strength of the CBCF composites was effectively improved, so that the compressive strength of the CBCF composites in the XY direction reached 0.30 MPa. Meanwhile, it is important that the Glu-PAM hydrogel was successfully applied as a binder for the CBCF composites, replacing the traditional resin binder, and the extrusion-injection molding method was used to achieve a high-efficiency, low-cost, and green preparation of the CBCF composites, which provides a new strategy for the preparation of CBCF composites.

## Figures and Tables

**Figure 1 nanomaterials-13-01045-f001:**
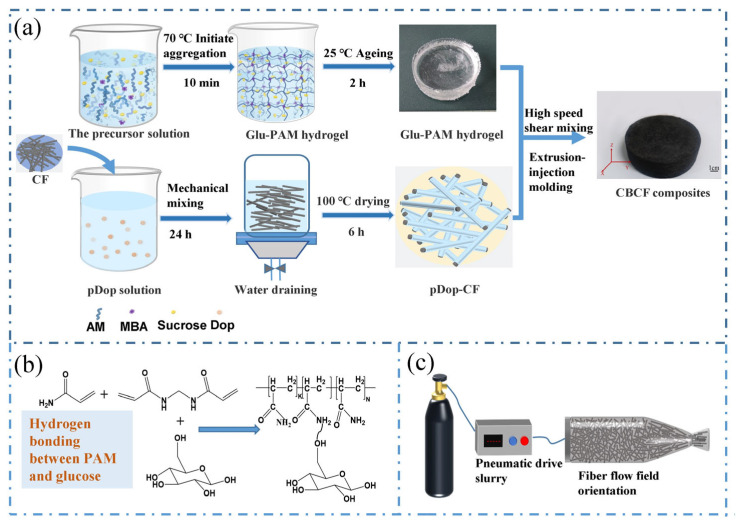
(**a**) Flow chart of CBCF slurry preparation; (**b**) reactions in the gelation process; (**c**) schematic diagram of the homemade extrusion molding equipment.

**Figure 2 nanomaterials-13-01045-f002:**
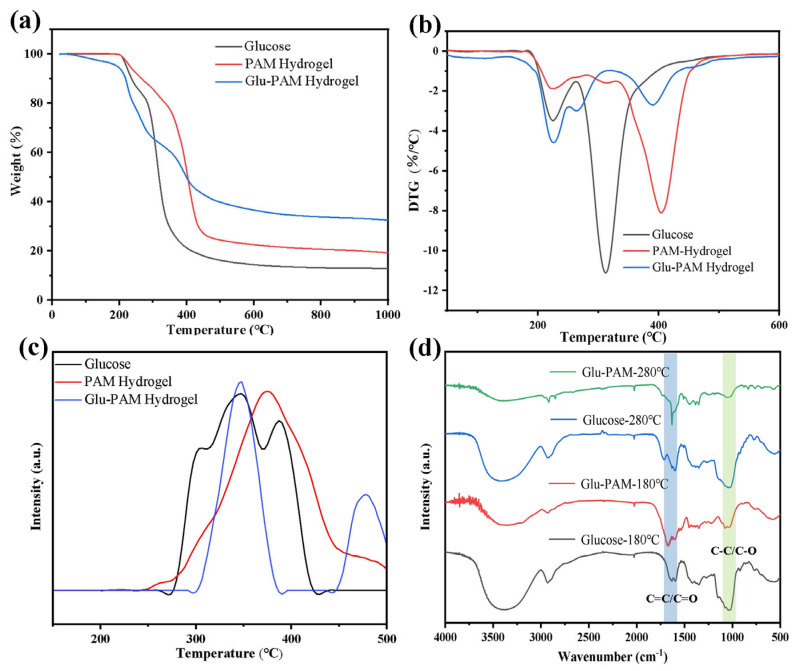
Thermal analysis and structural characterization of Glu-PAM hydrogel: (**a**) TG curve, (**b**) DTG curve, (**c**) TG-MS (CO_2_, *m*/*z* = 44) curve, and (**d**) FTIR curve.

**Figure 3 nanomaterials-13-01045-f003:**
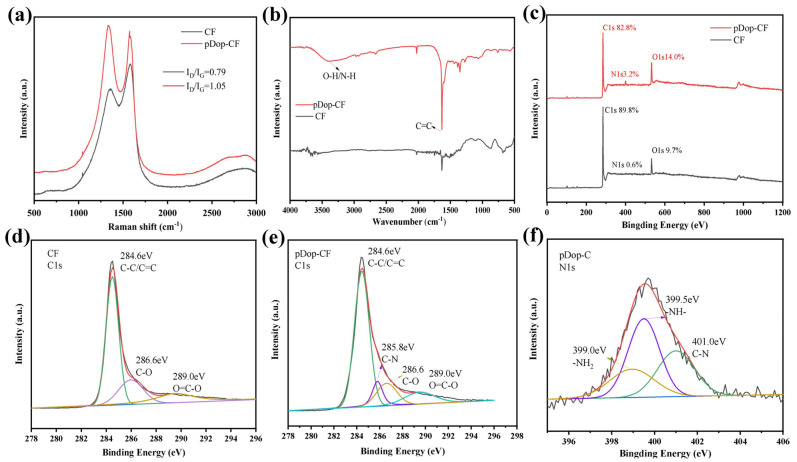
(**a**) Raman spectra, (**b**) FTIR spectra, and (**c**) XPS curves of CF and pDop-CF. (**d**) C1s profiles of CF, (**e**) C1s profiles of pDOP-CF, and (**f**) N1s profiles of pDOP-CF.

**Figure 4 nanomaterials-13-01045-f004:**
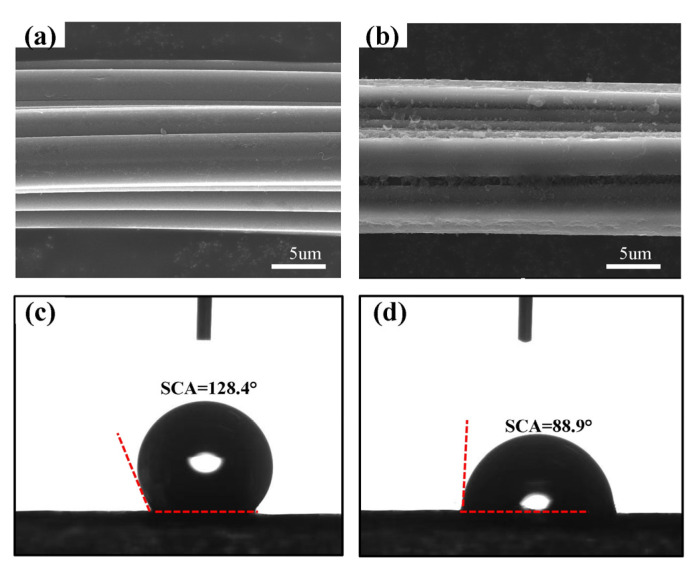
(**a**,**b**) SEM images; (**c**,**d**) static contact angles of CF and pDop-CF surfaces.

**Figure 5 nanomaterials-13-01045-f005:**
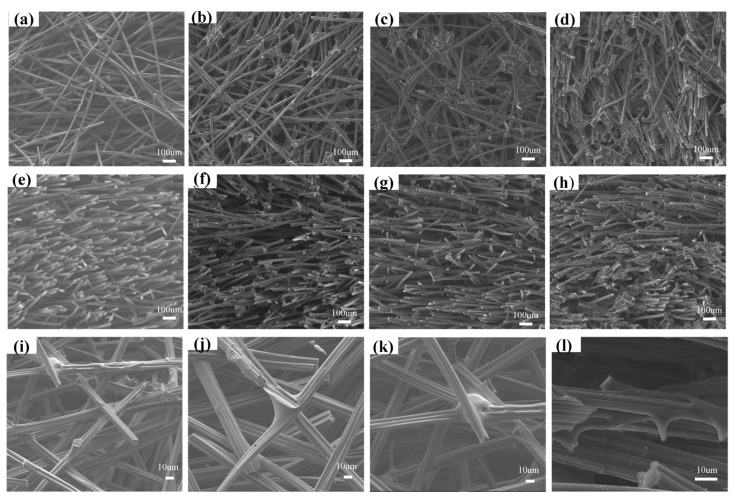
SEM images in XY direction and Z direction of (**a**,**e**) CBCF-1, (**b**,**f**) CBCF-2, (**c**,**g**) CBCF-3, and (**d**,**h**) CBCF-4. (**i**–**l**) High magnification SEM images of CBCF-1.

**Figure 6 nanomaterials-13-01045-f006:**
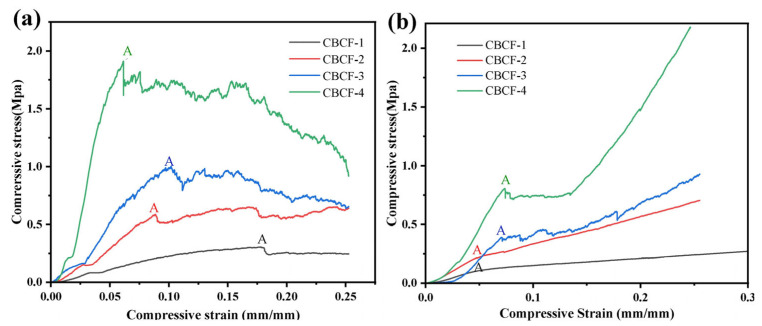
(**a**) XY direction and (**b**) Z direction stress–strain curves of CBCF-1, CBCF-2, CBCF-3, and CBCF-4.

**Figure 7 nanomaterials-13-01045-f007:**
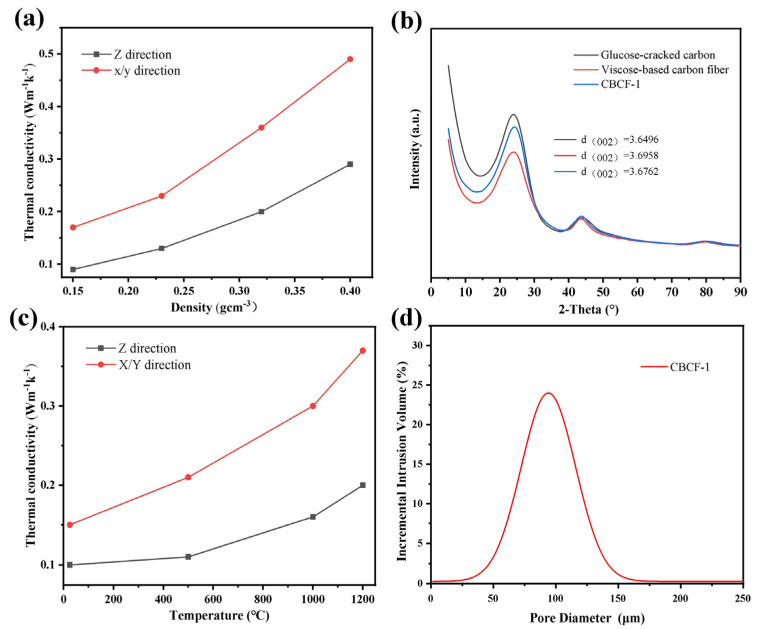
(**a**) Thermal conductivity of CBCF-1, CBCF-2, CBCF-3, and CBCF-4; (**b**) high temperature thermal conductivity of CBCF-1; (**c**) XRD patterns of glucose-cracked carbon; and (**d**) pore-size distribution of CBCF-1.

**Figure 8 nanomaterials-13-01045-f008:**
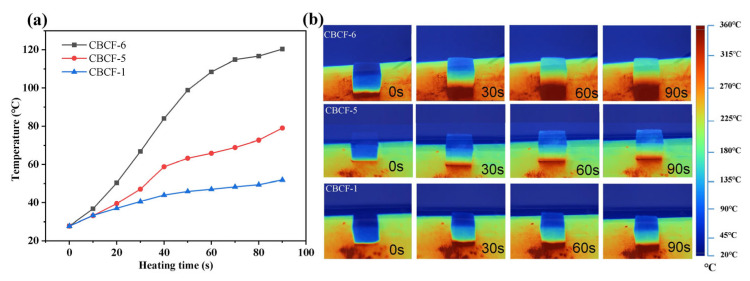
(**a**) Infrared thermal images of CBCF-1, CBCF-5, and CBCF-6; and (**b**) surface temperature of CBCF-1, CBCF-5, and CBCF-6 with different heating times.

## Data Availability

Data supporting the results of this study are available from the corresponding author.

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
