# Peer review of "Eco-Friendly Preparation of Carbon-Bonded Carbon Fiber Based on Glucose-Polyacrylamide Hydrogel Derived Carbon as Binder"

_nanomaterials, 2023, doi:10.3390/nano13061045_

Round 1
Reviewer 1 Report
In this work carbon fibers were modified with dopamine and mixed with a glucose-PAM hydrogel. After pyrolysis CBCF composites were obtained and characterized by various means. There is a large portion of experimental work and the results are adequately discussed. The conclusions are supported by the presented results. Below are some issues to be addressed.
1. The title is too long and should be revised.
2. Line 109. How the materials with different density were produced? By being exposed at 1200 oC for different time?
3. Lines 166-173. Please improve English in this text.
4. Lines 205-208. By this discussion it is implied that the ID/IG ratio decreased due to the decrease of ID. However, the intensity IG at 1580 cm-1 may increase after modification due to overlapping band of dopamine. The Raman and FTIR spectra of pure dopamine (either to be measured either from the literature) may shed light into this issue.
5. Line 275. Perhaps the elastic modulus along with the compressive strength will be more informative.
6. Line 307 The title of section 3.4 is repeated two times.
7. Line 346 Figure 7b is discussed after 7c and 7d. Thus the numbering of figures should be rearranged.
English syntax errors, typos etc
Line 20 “..could be were…”
Lines 180-185. There are a lot of full stops but no capital letter after the full stop.
Line 350 …the results are because.…
Author Response
Dear Reviewers
Please refer to the attachment.
Best regards,

Reviewer 2 Report
Recommendation: Minor Revision
1) The quality of the figure is low.
2) The authors treated the samples at 1200 °C for 2 h. Are the temperature and time optimized? What happens if a mild temperature (below 900 °C) is used?
3) The mechanical stability of the material can be examined by performing the folding test. The authors should provide optical images of the folding/bending test.
4) The introduction should also cover cellulose as a carbon precursor. The following references closely related to this topic may need to be added to the references.
- https://doi.org/10.1016/j.susmat.2022.e00450
Author Response
Dear Reviewers:
Please see the attachment.
Best regards
R.Tan
